# The Role of Maladaptive Plasticity in Modulating Pain Pressure Threshold Post-Spinal Cord Injury

**DOI:** 10.3390/healthcare13030247

**Published:** 2025-01-26

**Authors:** Marta Imamura, Rafaela Machado Filardi, Guilherme J. M. Lacerda, Kevin Pacheco-Barrios, Gilson Shinzato, Linamara Rizzo Battistella, Felipe Fregni

**Affiliations:** 1Instituto de Medicina Física e Reabilitação, Hospital das Clínicas HCFMUSP, Faculdade de Medicina, Universidade de São Paulo, São Paulo 04116-040, Brazil; marta.imamura@fm.usp.br (M.I.);; 2Neuromodulation Center and Center for Clinical Research Learning, Spaulding Rehabilitation Hospital and Massachusetts General Hospital, Harvard Medical School, Boston, MA 02138, USA; rafagoesmachado@gmail.com (R.M.F.);; 3Unidad de Investigación para la Generación y Síntesis de Evidenciasen Salud, Vicerrectorado de Investigación, Universidad San Ignacio de Loyola, Lima 150114, Peru; 4Departamento de Medicina Legal, Bioética, Medicina do Trabalho e Medicina Física e Reabilitação do da Faculdade de Medicina da Universidade de São Paulo (FMUSP), São Paulo 01246-903, Brazil

**Keywords:** central sensitization, spinal cord injury, pressure pain threshold, quantitative sensory test

## Abstract

Background: Spinal cord injury (SCI) frequently leads to pain, leading to significant disability. Pain sensitization, a key feature of SCI, is commonly assessed via quantitative sensory testing like the Pressure Pain Threshold (PPT), though the factors influencing PPT changes remain unclear. This study hypothesizes that specific clinical and neurophysiological factors modulate PPT in SCI patients. The primary objective is to identify predictors of PPT in SCI patients. Methods: We conducted a cross-sectional analysis of neurophysiological, clinical, and demographic data from 102 SCI patients in an ongoing prospective cohort study called “Deficit of Inhibition as a Marker of Neuroplasticity” (DEFINE study). Multivariable regression analyses were used to evaluate demographic, clinical, and functional variables associated with PPT, the primary outcome measure. Results: The sample comprised 87.9% males with an average age of 41. Trauma was the leading cause of SCI (77.45%), predominantly affecting the cervical and thoracic levels. Pain was reported by 44% of participants, and the mean PPT was 8.3 kPa, measured bilaterally. Multivariate analysis of PPT in the left, right, and bilateral thenar regions revealed consistent trends. Significant negative associations were found between bilateral PPT and low beta EEG frequency in the central area (β = −14.94, *p* = 0.017), traumatic lesion etiology (β = −1.99, *p* = 0.038), and incomplete lesions by the American Spinal Injury Association classification (β = −1.68, *p* = 0.012). In contrast, positive associations were observed with age (β = 0.08, *p* < 0.001). Conclusions: Our findings show that increased beta oscillations and traumatic brain injury having a lower PPT indicate that factors associated with maladaptive plasticity are associated with decreased and likely less functional PPT. On the other hand, increased motor function may help to regulate PPT in a more functional status.

## 1. Introduction

Spinal cord injury (SCI) is a debilitating neurological condition characterized by significant impairments in motor, sensory, and autonomic functions [1]. Over the past three decades, the global incidence of SCI has increased, with rates ranging from 236 to 1298 cases per million individuals across various countries [2]. In the United States alone, around 17,700 new cases are reported annually [3]. SCI is primarily caused by motor vehicle accidents (36–48%), violence (5–29%), falls (17–21%), and recreational activities (7–16%) [4]. The extent of functional impairment is categorized by the degree of preserved neurological function below the level of injury, as classified by the American Spinal Injury Association Impairment Scale (AIS) [3].

A significant consequence of SCI is chronic pain, which is associated with maladaptive neurophysiological and neurochemical changes in the somatosensory system [5]. Studies indicate that 30% to 80% of individuals with SCI experience persistent pain [5], and nearly one-third endure severe pain [6]. Pain sensitization in SCI is commonly assessed using the Pressure Pain Threshold (PPT), which measures the minimum pressure required to elicit pain. Lower PPTs are observed in individuals with chronic pain compared to pain-free individuals [7,8]. PPT is a non-invasive, easy-to-administer tool for studying chronic pain in conditions like SCI [7].

Several studies have explored PPT in SCI patients, identifying factors such as lesion location, motor strength, and psychological variables like anxiety as significant contributors to pain sensitization [9,10,11]. For example, Defrin et al. (2001) demonstrated that thermal and tactile sensations were more impaired in SCI patients with chronic pain, linking these deficits to peripheral and central sensitization mechanisms [11]. Additionally, individuals with incomplete SCI have been shown to experience greater pain sensitivity due to increased sensory hypersensitivity below the lesion level [9]. Motor strength, as assessed through handgrip tests, and psychological factors like anxiety have also been linked to altered PPT, with reduced motor function and heightened anxiety correlating with lower PPT [12].

While these studies provide valuable insights, they often focus on neuropathic pain and do not explore the multivariate relationship between clinical and neurophysiological factors in broader SCI populations. Our study aims to address this gap by examining clinical and neurophysiological predictors of pain sensitization, including SCI patients with and without chronic pain. This study hypothesizes that specific clinical and neurophysiological factors influence the modulation of PPT in patients with SCI.

## 2. Methods

### 2.1. Study Design

We conducted a cross-sectional analysis of patients with spinal cord injury (SCI) as part of an ongoing prospective cohort study called “Deficit of Inhibition as a Marker of Neuroplasticity” (DEFINE study) in rehabilitation [13]. For this cross-sectional study, we used only the assessment before the intervention. The protocol for the DEFINE study received approval from the Research and Ethical Committee of Hospital das Clínicas at the University of São Paulo School of Medicine (HC FMUSP) under registration number 86832518.7.0000.0068. All procedures and methods adhered to Brazilian research ethics guidelines and the Declaration of Helsinki.

### 2.2. Participants

Inclusion criteria: Participants were required to meet the following criteria: (1) were aged 18 or older, (2) were male or female, (3) had a clinical and radiological diagnosis of SCI (magnetic resonance imaging (MRI) or computerized tomography), (4) had clinical stability confirmed through medical evaluation, (5) had signed an informed consent form, and (6) met the eligibility requirements for the Instituto de Medicina Física e Reabilitação (IMREA).

Exclusion criteria: Participants were excluded if they (1) were pregnant or (2) had any clinical or social conditions that might interfere with participation in the rehabilitation program. Patients with adequate functionality who had specific comorbidities, such as hypertension or diabetes, were not excluded.

### 2.3. Study Procedures

Patients with SCI who were enrolled in the IMREA’s conventional rehabilitation program were invited to join this study and were included upon signing the informed consent form. A qualified researcher conducted various clinical and neurophysiological assessments during a single visit. The tools used were chosen to facilitate a comprehensive evaluation of the patients. All assessments were performed by the same examiner, who was trained to ensure standardized evaluations. One of the assessments was used as the dependent variable for this study, namely, Pressure Pain Threshold (PPT), as we aim to understand pain sensitivity in this specific population better.

### 2.4. Clinical and Functional Assessments

Demographic and baseline clinical information was extracted through a standardized medical interview, incorporating details about age, gender, race, marital status, body mass index, employment status, alcohol consumption or smoking, chronic health conditions, and long-term medications. The patient’s disease history of SCI was collected, including the time of lesion, time of hospitalization, etiology of the lesion, type of incapacity, level of the lesion, and AIS score. Motor Strength Function was evaluated using various tests, including the Handgrip Strength Test, Pinch Strength Test, Purdue Pegboard Test, Walking Index for Spinal Cord Injury, and Medical Research Council Scale. Additionally, to thoroughly characterize this study’s sample, we conducted a multidimensional assessment covering several domains: pain (Visual Analog Scale, Pressure Pain Threshold, and Conditioned Pain Modulation), cognitive function (Montreal Cognitive Assessment), level of independence (Functional Independence Measure), and emotional well-being (Hospital Anxiety and Depression Scale). For a detailed explanation of all other assessments conducted, please consult our (reliability) published protocol [13]. See the diagram of assessments in Figure 1.

#### 2.4.1. Pressure Pain Threshold (PPT)

PPT was defined as the lowest pressure level that causes pain, as measured with an algometer and recorded in kilopascals (kPa) [8]. We took three readings with 15 s intervals between each, and the average was used as the final PPT value. Measurements were taken in the thenar region on the right hand and the left hand. The PPT bilateral is the average of the PPT left and right.

#### 2.4.2. American Spinal Injury Association Impairment Scale (AIS)

This scale assesses the severity of spinal cord injuries using a 5-point system ranging from A (complete injury) to E (normal function). The final score includes the sum of sensory and motor scores, in addition to sensory and motor anal assessment. For the analysis, patients classified as A were categorized as having a complete lesion, while those classified as B, C, D, or E were categorized as having an incomplete lesion.

#### 2.4.3. Hospital Anxiety and Depression Scale (HADS)

This is a 14-item assessment designed to both quantify and qualify symptoms related to anxiety and depression. The scale includes 14 multiple-choice items divided into two subscales, one for anxiety and another for depression, each containing seven items. The overall score on each subscale ranges from 0 to 21. The tool is specifically aimed at identifying mild forms of affective disorders in non-psychiatric settings. Patients are instructed to respond based on how they have felt over the past week.

#### 2.4.4. Handgrip Strength Test

The Handgrip Strength Test is designed to assess muscle strength using a continuous measurement on a kilogram-force scale, measured explicitly with a hand grip dynamometer. This test evaluates the maximum force exerted during a five-finger squeeze. To determine the average strength, three trials are performed on each hand. During the test, participants sit with their arms close to the torso, elbows bent at a 90-degree angle, forearms in a neutral position, and wrists slightly extended. The Handgrip Strength Test bilateral was determined by calculating the average of the means from both the right and left sides.

#### 2.4.5. Sensory Vibration Test

A validated way of measuring profound/deep sensitivity is tested on the bone prominences on the right and left upper limbs.

#### 2.4.6. Electroencephalography (EEG)

The EEG session lasted around 45 min, with 25 min devoted to preparing the participant and configuring the software. This was followed by 10 min of EEG recording, which included a 5 min resting phase with eyes open and another 5 min with eyes closed. During the resting phase, participants were asked to stay relaxed while the investigator ensured they did not fall asleep. The task-related segment, which took 8 min, involved activities such as movement observation, imagery, and execution, all recorded using ANT Neuro 64-channel EEG system (eego™ Software by ANT Neuro) connected to E-Prime. This segment included 60 trials, with 20 trials for each type of movement task presented randomly. Following standard procedures, the EEG data were collected using a 64-channel EEG system (ATN Neuro, Enschede, The Netherlands). The data were filtered with a band-pass range of 0.3−200 Hz and digitized at a sampling rate of 250 Hz. On average, 1% of trials were excluded due to artifacts, with exclusions ranging from 0 to 4%. A clinical neurophysiologist reviewed the EEG data to identify artifacts and potential clinical abnormalities. The processed data were then exported for further analysis using MATLAB (R2014b, The MathWorks Inc., Natick, MA, USA) and EEGLab. The analysis focused on the following frequency bands: delta (2−4 Hz), theta (4−8 Hz), low alpha (8−10.5 Hz), high alpha (10.5−13 Hz), alpha (8−13 Hz), low beta 1 (13−20 Hz), high beta 2 (20−30 Hz), and beta (13−30 Hz).

### 2.5. Statistical Analysis

We described the baseline characteristics of the sample using descriptive statistics. Categorical variables were presented with absolute and relative frequencies (percentages), while continuous variables were summarized using means and standard deviations (SD). We aimed to examine the relationship between demographic and clinical factors and pain thresholds in thenar regions.

We conducted univariate linear regressions using the thenar region’s PPTs as outcome variables and assessed relevant demographic, cognitive, emotional, and motor function variables as predictors. Variables with a *p*-value of ≤0.20 were initially considered for inclusion in further modeling. Following the existing literature, we decided beforehand to include specific covariates in the multivariate analysis regardless of their significance in the univariate regression analysis. These covariates included sex, body mass index (BMI), age, pain intensity, depression, lesion level, type of incapacity, and motor function. Three distinct multivariate regression models were created to analyze the pain thresholds for the thenar regions: one for the PPT on the left side, one for the PPT on the right side, and one for the averaged bilateral PPT values.

In our modeling process, variables significantly associated with the outcome or that acted as confounders were retained in the model. A confounder was defined as a variable that altered the beta coefficient of another independent variable by more than 10% and was theoretically related to both the outcome and the predictor. The best-fitting model was identified based on the adjusted R-squared value. We assumed normality based on the central limit theorem and verified it through visual inspection of the data distributions. The normality of the residuals was assessed using QQ plots and the Shapiro–Wilk test. We visually inspected residuals plotted against predicted values to verify the assumption of homoscedasticity. In this exploratory analysis, we did not apply corrections for multiple comparisons to reduce the risk of type II errors. However, we limited the number of comparisons by focusing on a predefined list of significant covariates (forced independent variables).

We employed the Sobol method for global sensitivity analysis, which involves decomposing the variance of the model output to attribute it to different input variables. Input variables were varied across their entire plausible range, and their effects on the output were quantified. The Sobol indices were calculated to measure the contribution of each variable to the output variance. The sensitivity analysis was tested for the dependent variable PPT bilateral, and the study focused on the independent variables present in the multivariate analysis model.

All analyses were performed using R Studio 4.1.1 (R Foundation for Statistical Computing, Vienna, Austria). *p*-values less than 0.05 were considered statistically significant.

## 3. Results

### 3.1. Sample Characteristics

Our sample consisted of 102 participants with SCI, with the majority being male (87.9%) and an average age of 41 years (±16). The leading cause of the injury was trauma (77.45%), with the cervical (47.06%) and thoracic (40.21%) levels being the most affected. The symptom of pain was present in 44% of the sample, and the mean PPT in the thenar region was 8 kPa (±3.06) on the left side and 8.5 kPa (±3.01) on the right side, with an overall average of 8.3 kPa (±3) for both sides. Sample characteristics are summarized in Table 1 and the Appendix A.

### 3.2. Dependent Variable Characteristics

We analyzed 91 participants, with 11 (10.78%) unable to complete the procedure. The PPT showed a normal distribution on the right and left sides and an average of the sides for the bilateral measurement. The median PPT for the left hand was 8.17 kPa (95% CI: 7.66–8.86), with an IQR of 6.17–10.42. For the right hand, the median PPT was 8.32 kPa (95% CI: 7.98–9.21), and the IQR was 6.15–10.71. The bilateral PPT had a median of 8.49 kPa (95% CI: 7.83–8.99), with an IQR of 6.28–10.27. These results are summarized in Table 2 and the Appendix A.

### 3.3. Univariate Analysis

#### 3.3.1. PPT on the Left Side

The univariate analysis of PPT left side identified several significant covariates, including level of SCI (non-cervical, β = 2.4, *p* < 0.001), Motor Strength Function (as indexed by Handgrip Strength Test in the left hand (β = 0.07, *p* < 0.001) and Pinch Strength Test in the left hand (β = 0.2, *p* = 0.045)), AIS (incomplete lesion, β = −2.03, *p* = 0.002), Sensory Vibration Test in the superior member of the left side (β = 4.25, *p* = 0.019), and years of education (β = −0.14, *p* = 0.035). The EEG identified several significant covariates, particularly in the low beta frequency, with notable findings in the frontal region (β = −16.81, *p* = 0.012), parietal region (β = −14.85, *p* = 0.022), and central region (β = −12.06, *p* = 0.04). Further details are provided in the Appendix A.

#### 3.3.2. PPT on the Right Side

The univariate analysis of the PPT on the right side revealed several significant variables, including the AIS (incomplete lesions, β = −2.28, *p* < 0.001), level of SCI (non-cervical, β = 1.99, *p* < 0.001), Motor Strength Function (as indexed by Handgrip Strength Test in the right hand (β = 0.05, *p* = 0.003)), and years of education (β = −0.16, *p* = 0.027). Further details are provided in the Appendix A.

#### 3.3.3. PPT Bilateral

The univariate analysis of the averaged bilateral PPT values identified several significant variables: level of SCI (non-cervical, β = 2.21, *p* < 0.001), Motor Strength Function (as indexed by the average Handgrip Strength Test across both sides (β = 0.06, *p* < 0.001)), AIS (incomplete lesions, β = −2.18, *p* < 0.001), years of education (β = −0.15, *p* = 0.023), and EEG low beta frequency in the frontal (β = −13.48, *p* = 0.036) and parietal (β = −12.58, *p* = 0.044) regions across both sides. Further details are provided in the Appendix A.

### 3.4. Multivariate Analysis

#### 3.4.1. PPT on the Left Side

In the multivariate analysis (Table 3), traumatic lesion etiology was negatively associated with PPT (β = −2.04, *p* = 0.046), as were incomplete lesions based on AIS classification (β = −1.45, *p* = 0.042). These findings show that both traumatic lesions and incomplete SCI are linked to increased pain sensitivity, reflected in lower PPT values. In contrast, the Sensory Vibration Test on the left side (β = 3.99, *p* = 0.018), age (β = 0.05, *p* = 0.045), motor strength (as indexed as Handgrip Strength Test in the left hand (β = 0.09, *p* < 0.001)), and the Hospital Anxiety Scale (β = 0.17, *p* = 0.04) showed positive associations with PPT. These results indicate that increased vibratory sensation, age, Motor Strength Function, and anxiety levels are linked to higher PPT. The analysis was adjusted for sex. The multivariable model demonstrated an adjusted R-squared of 0.38 (*p* < 0.001).

#### 3.4.2. PPT on the Right Side

In the multivariate analysis (Table 4), the EEG low beta frequency in the central region of the right side showed a strong negative association with PPT (β = −17.65, *p* = 0.02), as did traumatic lesion etiology (β = −2.33, *p* = 0.046) and incomplete lesions based on AIS classification (β = −2.15, *p* = 0.005). This indicates these factors are linked to lower PPT and increased pain sensitivity. Conversely, age (β = 0.1, *p* < 0.001) and the Motor Strength Function (as indexed as Handgrip Strength Test in the right hand (β = 0.047, *p* = 0.02)) were positively associated with PPT, indicating that older age and greater Motor Strength Function are linked to higher PPT and reduced pain sensitivity. The analysis was adjusted for sex. The multivariable model demonstrated an adjusted R-squared of 0.33 (*p* < 0.001).

#### 3.4.3. PPT Bilateral

In the multivariate analysis (Table 5), a strong negative association was found between the EEG low beta frequency in the central region and PPT (β = −14.94, *p* = 0.017), as well as with traumatic lesion etiology (β = −1.99, *p* = 0.038) and the incomplete lesions based on AIS classification (β = −1.68, *p* = 0.012), indicating that these factors are linked to lower PPT and heightened pain sensitivity. In contrast, age (β = 0.08, *p* < 0.001) and Motor Strength Function (as indexed by the average Handgrip Strength Test across both sides (β = 0.065, *p* < 0.001)) were positively associated with PPT, indicating that older age and greater overall Motor Strength Function are linked to higher PPT and reduced pain sensitivity. The analysis was adjusted for sex. The multivariable model had an adjusted R-squared of 0.40 (*p* < 0.001).

### 3.5. Sensitivity Analysis

The sensitivity analysis of the independent variables present in the multivariate analysis model for PPT bilateral was tested. We used the Sobol Sensitivity Analysis, a global sensitivity analysis method used to quantify the contribution of input variables to the output variance of a mathematical model [14]. In Sobol Sensitivity Analysis, the first-order Sobol index (Si) is used to assess the influence of input parameters on the model’s output. When the Si is close to 1, the parameter has a strong independent effect on the output, meaning it explains a significant portion of the variance [14]. Conversely, when the Si is close to 0, the parameter has little or no independent influence on the outcome.

The age, sex, etiology of lesion, AIS, and EEG show relatively lower Sobol indices, meaning they have less impact on the output of the model, showing the independent contribution of each input. The Motor Strength Function (as indexed by Handgrip Strength Test bilateral) stands out with a higher Sobol index, near 0.9, indicating that it strongly influences the model’s output. Further details are provided in the Appendix A.

## 4. Discussion

### 4.1. Current Understanding of PPT in Individuals Affected by SCI

The main finding of this study was an exploratory relationship between PPT and clinical findings, demographics, cognitive function, and motor function among the patients affected by SCI. Pain perception in individuals with SCI is complex and varies significantly depending on the level and severity of the injury. After SCI, patients often experience a range of pain types, including neuropathic pain, which can be particularly challenging to manage due to its resistance to conventional analgesics [15]. Neuropathic pain, which arises from damage to the somatosensory nervous system, is categorized in SCI into three specific types: pain at the level of injury, pain occurring below the level of injury, and other types of neuropathic pain [16]. Assessing pain in individuals with SCI is a complex process that requires not only standard pain evaluation tools but also a specialized assessment [15]. In this evaluation, using PPT as an assessment can help understand the pain sensitization in this population.

### 4.2. PPT and Age

In this study, we observed that increasing age is slightly associated with a higher PPT in patients with SCI, which contrasts with the expected inverse relationship between age and PPT in the general healthy population. Typically, in non-SCI populations, age-related changes in the nervous system, such as a decline in pain modulation efficiency, lead to greater pain sensitivity and lower PPT values [17,18,19,20,21]. A meta-analysis by Lautenbacher et al. (2017) indicates that aging is typically associated with increased pain sensitivity due to changes in pain modulation systems [22].

However, our study observed that increasing age is slightly associated with higher PPT in patients with SCI. This counterintuitive finding may reflect unique age-related changes in sensory processing and neuroplasticity specific to SCI. Older individuals with SCI may experience a reduction in peripheral sensory input due to age-related degeneration of small-diameter nerve fibers [19]. This diminished nociceptive input could partially account for higher PPT in older patients, as the central nervous system receives less afferent signaling from damaged peripheral pathways. Neuroplasticity also plays a crucial role in shaping pain perception in SCI patients. Prolonged exposure to chronic pain can induce long-term adaptations in the central nervous system, including alterations in the balance between excitatory and inhibitory signaling. Older SCI patients may demonstrate enhanced activation of descending inhibitory pathways, mediated by structures such as the periaqueductal gray (PAG) and rostral ventromedial medulla [23]. These mechanisms could contribute to increased PPT by dampening pain signals at the spinal or supraspinal level.

Additionally, this finding can be explained by the age distribution of our sample, which includes a higher proportion of younger individuals. Younger people tend to exhibit more heightened or reactive responses to pain stimuli, often referred to as “overreactive” pain sensitivity [24]. This could be due to their nervous system’s heightened reactivity, as they have not yet developed the long-term adaptive mechanisms that older individuals with chronic pain may acquire over time [25]. In younger patients, the nervous system may still be in a more acute phase of reacting to pain, contributing to lower PPT [26]. These results emphasize the need to consider the unique characteristics of the SCI population when analyzing pain sensitivity across different age groups.

### 4.3. PPT and Relation with the Characteristics of the SCI

Pressure Pain Threshold (PPT) in patients with spinal cord injury (SCI) is significantly influenced by the characteristics of the injury, notably classified according to the American Spinal Injury Association (AIS) Scale [27]. Our findings indicate that patients with incomplete SCIs, categorized as AIS grades B through E, exhibit increased pain sensitivity (reflected by lower PPT). This suggests that even when some sensory and motor functions remain intact, these incomplete lesions’ altered neural processing of pain signals leads to heightened sensitivity [28]. Due to partial nerve pathway preservation, the interaction between undamaged and damaged neural circuits likely exacerbates pain responses in these individuals [29]. Similar phenomena, showing increased pain sensitivity in patients with incomplete SCI, have been consistently reported in prior studies [30,31].

Furthermore, the etiology of SCI also plays a crucial role in determining PPT outcomes. Our analysis reveals that non-traumatic SCIs are associated with significantly lower PPTs compared to traumatic injuries. This difference may be attributed to the chronic and progressive nature of non-traumatic conditions, which typically involve gradual onset and continuous, low-level nerve irritation [32]. Over time, this prolonged exposure to pathologic processes may lead to extensive maladaptive neural changes, such as central sensitization, thus amplifying pain perception and further reducing PPT [33]. Supporting this, Bresnahan (2022) observed heightened pain sensitivity in non-traumatic SCI cases, underscoring the impact of injury etiology on pain sensitivity [34].

### 4.4. PPT and Motor Function

Our study suggests a slight positive association between the Handgrip Strength Test and PPT. The Handgrip Strength Test is a dynamometric assessment that measures the maximum force exerted during a grip, indicating overall hand and forearm muscle strength [35]. The Handgrip Strength Test might be indirectly related to pain sensitivity. Strong grip strength could be associated with better overall muscle function and possibly a higher tolerance to pressure-induced pain [36,37]. However, the small estimate indicates that this effect is modest and may not be the primary determinant of PPT.

### 4.5. PPT and Sensitive Function

In our model, the findings on the left side suggest that the Sensory Vibration Test may be associated with PPT, as both assess sensory and pain responses. The Sensory Vibration Test, commonly used to evaluate vibratory sensation, might relate indirectly to PPT by providing insights into the integrity of sensory pathways [38]. Areas with intact vibratory sensation, as detected by the Sensory Vibration Test, may also demonstrate higher pain tolerance, reflected in a higher PPT. This suggests a potential link between preserved vibratory function and increased resistance to pain, as these intact sensory pathways might play a crucial role in more effectively modulating pain perception [39,40,41].

### 4.6. PPT and Cognition Assessment

Anxiety, depression, substance disorders, and post-traumatic syndrome disorder are markedly more common among individuals with SCI compared to the general population [42]. Our study indicates a positive association between anxiety levels and pain tolerance, implying that higher anxiety might be linked to a slight increase in pain threshold. This finding challenges the conventional view that anxiety typically decreases pain tolerance, as it tends to increase the perception of pain and distress [43].

However, this relationship can be more complex in specific contexts, such as in SCI. SCI often leads to profound changes in the autonomic nervous system and central pain modulation pathways, which can contribute to altered sensory processing. Stress-induced analgesia (SIA) is a phenomenon wherein exposure to stressors, such as anxiety, triggers the activation of the body’s endogenous pain inhibition systems [44,45]. This mechanism is mediated primarily by the release of endogenous opioids and other neurotransmitters, such as norepinephrine and serotonin, which act on descending inhibitory pathways to suppress pain perception [46]. In SCI patients, these mechanisms may be further altered due to neuroplastic changes in the central nervous system. Disruption of the usual pain transmission pathways, combined with increased reliance on descending inhibitory control, may result in atypical responses to anxiety, such as an elevated PPT [9]. This adaptation might serve as a compensatory survival mechanism against the chronic pain and discomfort often experienced in this population. This activation of SIA can temporarily increase the PPT, effectively reducing pain perception during stressful situations. This mechanism is a key example of how the body adapts to chronic stress or pain by enhancing pain tolerance as a survival strategy [47].

### 4.7. PPT and EEG

Our study suggests a significant negative association between low beta intensity in the central region and PPT. EEG low beta activity (ranging from 13 to 20 Hz) is often associated with active thinking, focus, and alertness. In the context of pain, heightened beta activity may indicate increased cognitive processing related to the experience of pain or anticipation of pain [48], likely also representing maladaptive disrupted activity [49]. The central region of the brain, particularly areas such as the somatosensory cortex, is crucial for processing sensory information, including pain (51). Our findings are consistent with the existing literature, demonstrating a correlation between low beta band activity and lower PPT, indicating increased pain sensitivity. Wang et al. (2024) analyzed EEG signals from SCI patients. They found that those experiencing neuropathic pain exhibited significantly higher beta band activity compared to the control group [50]. Another study by Simis (2021) on patients affected by SCI showed that worse clinical function was related to increased beta activity in the EEG [51]. Understanding this relationship is crucial for developing interventions that modulate beta activity to manage pain more effectively.

### 4.8. Limitations

This study has several limitations and should be viewed as exploratory. A key limitation is its cross-sectional design, which does not allow for an in-depth examination of changes in pain sensitivity over time. Issues such as variations in pain intensity and the delayed onset of neuropathic pain after SCI would be better addressed in a longitudinal study. However, a cross-sectional approach minimizes the risk of participant dropout and effectively maps the prevalence of altered PPT in the SCI population. The predominance of male participants in our sample, reflective of the real-world epidemiology of SCI, limits the generalizability of findings to female SCI patients. Another limitation of this study is that we only used mechanical stimuli for PPT testing with an algometer and did not include any thermal sensation assessments. In conjunction with the fact that PPT testing was conducted at the same site (the thenar region) for all patients, regardless of injury level or completeness, this restricts the comprehensiveness of our findings. For future research, it would be beneficial to incorporate multiple types of test stimuli, including thermal modalities, and to tailor the testing locations according to the specific injury level and sensory loss. This approach would enhance the significance of the results and potentially reveal new correlations, leading to a more nuanced understanding of pain sensitivity in individuals with spinal cord injuries.

To gain a more comprehensive understanding of pain sensitivity in SCI patients, more extensive studies should consider the influence of time by including a balanced representation of both early and late injuries, as well as both traumatic and non-traumatic cases.

## 5. Conclusions

Our study highlights the complex relationship between pain perception and various clinical, cognitive, and motor factors in SCI patients. Age is positively associated with PPT, contrasting with trends seen in the general population. Incomplete lesions based on AIS classification and non-traumatic etiologies were linked to increased pain sensitivity (lower PPT). Motor Strength and Sensory Function were modestly associated with higher PPT. Interestingly, anxiety demonstrated a positive relationship with PPT, suggesting adaptive mechanisms like stress-induced analgesia. Lastly, increased low beta EEG activity in the central region correlated with lower PPT, indicating heightened pain sensitivity. These findings emphasize the need for personalized pain management strategies in SCI patients, and further research should aim to confirm these findings through longitudinal studies and more comprehensive pain assessments.

## Figures and Tables

**Figure 1 healthcare-13-00247-f001:**
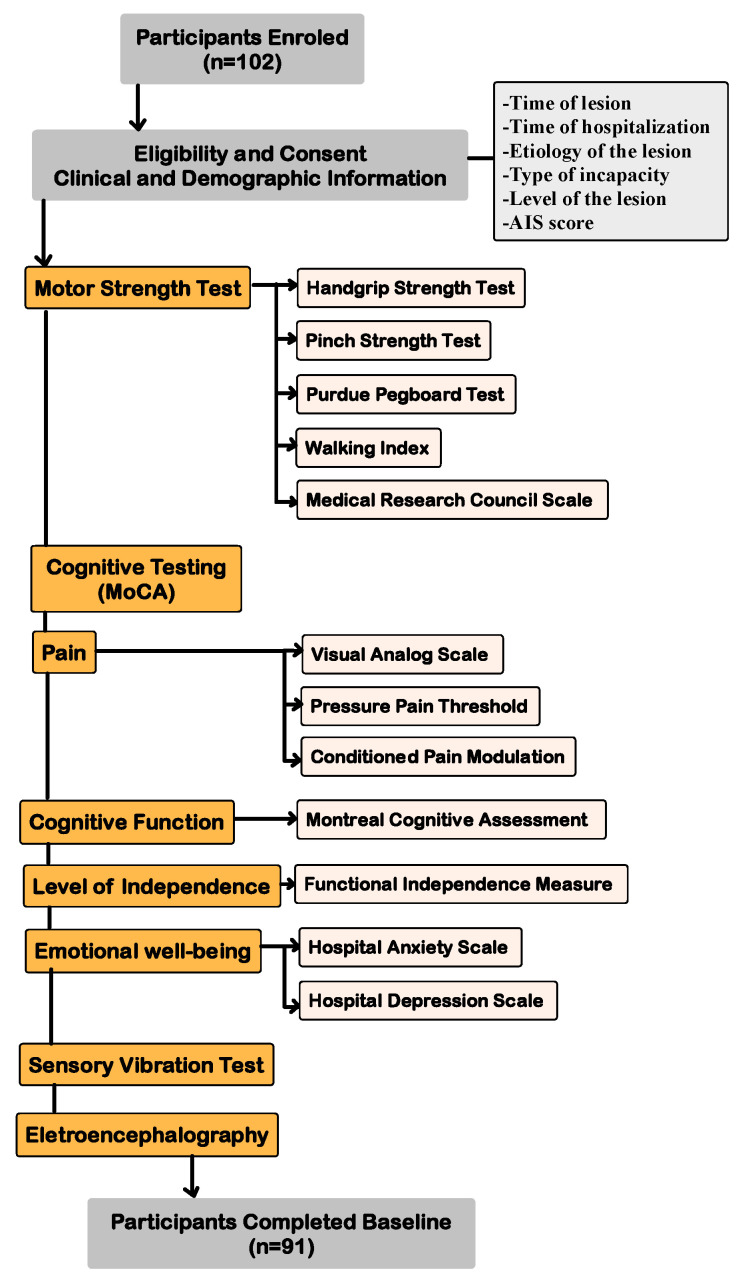
Diagram of assessments performed.

**Table 1 healthcare-13-00247-t001:** Baseline demographic and clinical characteristics.

	Participants, n = 102
Age (years)	41 (16)
Gender (%)	
Male	80 (87.9%)
Female	11 (12.1%)
Race	
White	41 (45.1%)
Black	10 (11%)
Mixed race	39 (42.9%)
Indigen	1 (1.1%)
Body mass index (kg/m^2^)	24.52 (5.01)
Time of lesion (months)	19.1 (23.1)
Time of hospitalization (days)	25.53 (7.2)
Etiology of lesion	
Traumatic	79 (77.45%)
Vascular	4 (3.92%)
Others	19 (18.63%)
Level of lesion	
Cervical	48 (47.06%)
Thoracic	41 (40.21%)
Lumbar	12 (11.76)
Sacral	1 (0.97)
Pain Pressure Threshold (kPa)	
Thenar region (left)	8 (3.06)
Thenar region (right)	8.53 (3.01)
Conditioned pain modulation (kPa)	
Thenar region (left)	2.18 (0.04)
Thenar region (right)	2.07 (0.14)
Hand Grip Test (kPa)	
Left side	32.56 (18.03)
Right side	32.68 (17.26)
Visual Analog Scale (VAS)	
Left side	2.27 (2.95)
Right side	2.05 (2.67)
Walking Index for Spinal Cord Injury	4.98 (7.25)
Hospital Anxiety Scale	4.95 (3.97)
Hospital Depression Scale	3.98 (3.19)
Functional Independence Measure	83.21 (23.7)
Montreal Cognitive Assessment	22.84 (4.45)
kPa: Kilopascal	
Continuous variables: mean (SD); categorical variables: n (percentage).	

**Table 2 healthcare-13-00247-t002:** Dependent variables characteristics.

Variable	Median (95% CI)	IQR	Missing Data: Number (%)
PPT left hand (kPa)	8.17 (7.66–8.86)	6.17–10.42	11 (10.78%)
PPT right hand (kPa)	8.32 (7.98–9.21)	6.15–10.71	11 (10.78%)
PPT bilateral (kPa)	8.49 (7.83–8.99)	6.28–10.27	11 (10.78%)
CI: confidence interval; IQR: interquartile range.			

**Table 3 healthcare-13-00247-t003:** Multivariable model of PPT left thenar region.

	Adjusted R-Squared, 0.38		
Variable	β-Coefficient	*p*-Value	95% Confidence Interval
Intercept	4.13	0.0046	1.357 to 6.913
Age (years)	0.049	0.0451	0.001 to 0.096
Etiology of lesion, non-traumatic	−2036	0.0464	−4.038 to −0.034
Asia Impairment Scale incomplete lesion (B, C, D, E)	−1446	0.0418	−2.835 to −0.057
Hand Grip Test on the left side	0.088	<0.001	0.049 to 0.127
Sensory Vibration Test in left upper limb	3992	0.0167	0.763 to 7.222
Hospital Anxiety Scale	0.166	0.041	0.007 to 0.324
Adjusted by sex.			

**Table 4 healthcare-13-00247-t004:** Multivariable model of PPT right thenar region.

	Adjusted R-Squared, 0.33		
Variable	β-Coefficient	*p*-Value	95% Confidence Interval
Intercept	6791	<0.001	3.985 to 9.595
Age (years)	0.105	<0.001	0.05 to 0.159
Etiology of lesion, non-traumatic	−2329	0.0468	−4.623 to −0.034
Asia Impairment Scale incomplete lesion (B, C, D, E)	−2.157	0.005	−3.627 to −0.687
Hand Grip Test on the right side	0.047	0.02	0.007 to 0.086
EEG central region left side and low beta frequency	−17,652	0.014	−31.621 to −3.683
Adjusted by sex.			

**Table 5 healthcare-13-00247-t005:** Multivariable model of PPT bilateral of thenar region.

	Adjusted R-Squared, 0.4		
Variable	β-Coefficient	*p*-Value	95% Confidence Interval
Intercept	6309	<0.001	3.787 to 8.832
Age (years)	0.083	<0.001	0.036 to 0.131
Etiology of lesion, non-traumatic	−1.996	0.039	−3.884 to −0.106
Asia Impairment Scale incomplete lesion (B, C, D, E)	−1681	0.012	−2.972 to −0.390
Hand Grip Test average bilateral	0.065	<0.001	0.03 to 0.101
EEG central region bilateral and low beta frequency	−14,943	0.017	−27.129 to −2.755
Adjusted by sex.			

## Data Availability

The original contributions presented in this study are included in the article/Appendix A. Further inquiries can be directed to the corresponding author.

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
