# Peer review of "The Role of Maladaptive Plasticity in Modulating Pain Pressure Threshold Post-Spinal Cord Injury"

_healthcare, 2025, doi:10.3390/healthcare13030247_

Round 1
Reviewer 1 Report
Comments and Suggestions for Authors
This cross-sectional study analyzed factors influencing pressure pain threshold (PPT) in 102 spinal cord injury (SCI) patients, utilizing baseline data from the ongoing “Deficit of Inhibition as a Marker of Neuroplasticity” (DEFINE) study in rehabilitation. The study highlights the complex relationship between pain perception and various clinical, cognitive, and motor factors in SCI patients. Age is positively associated with PPT, contrasting with trends seen in the general population. Incomplete lesions based on AIS classification and non-traumatic etiologies were linked to increased pain sensitivity (lower PPT). In comparison, motor strength function and sensory function were modestly associated with higher PPT. Interestingly, anxiety demonstrated a positive relationship with PPT, suggesting adaptive mechanisms like stress-induced analgesia. Lastly, increased low-beta EEG activity in the central region correlated with lower PPT, indicating heightened pain sensitivity. These findings emphasize the need for personalized pain management strategies in SCI patients.
While the research provides valuable insights into chronic pain management post-SCI, several critical aspects require improvement to enhance clarity, depth, and impact.
Major Critiques and Suggestions:
1. The abstract and introduction fail to present a clearly defined hypothesis or research objective. This omission diminishes initial engagement and leaves readers unclear about the study's primary goals.
-Explicitly state the hypothesis and research objectives in both the abstract and the introduction. For example, include phrases such as: “This study hypothesizes that specific clinical and neurophysiological factors modulate PPT in SCI patients.” This will guide readers and improve the study’s framing.
2. Results: The results section is overly text-heavy, making it difficult for readers to grasp the key findings and their significance. Incorporate graphical representations of the data to improve accessibility and interpretation. Suggested including plots showing correlations, such as PPT vs. age and PPT vs. motor strength.
3. Discussion: The study notes a positive association between age and PPT, contrasting with trends observed in the general population, but does not provide a physiological explanation for this observation.
- Expand the discussion to include potential mechanisms underlying this finding. For example, consider discussing age-related changes in sensory processing, neuroplasticity, or central sensitization. Reference relevant literature to provide a more nuanced understanding of this counterintuitive result.
4. The relationship between anxiety and increased PPT is noted but not adequately explained. The discussion does not delve into how stress-induced analgesia might modulate sensory processing in SCI patients.
- Provide a detailed explanation of stress-induced analgesia, supported by references to mechanisms such as endogenous opioid release or alterations in central pain modulation pathways. Discuss how these mechanisms might differ in SCI patients compared to the general population.
Minor Points
• The abstract does not define the ‘DEFINE’ cohort, potentially confusing readers unfamiliar with the study.
• Methods Line 113: “Motor Strength Function function” includes a typographical error.
• Expand the discussion to connect findings to broader neurological and clinical contexts.
• Highlight the clinical implications of these findings by outlining how they could inform personalized pain management strategies.
Author Response
REVIEWER 01:
While the research provides valuable insights into chronic pain management post-SCI, several critical aspects require improvement to enhance clarity, depth, and impact.
Major Critiques and Suggestions:
- The abstract and introduction fail to present a clearly defined hypothesis or research objective. This omission diminishes initial engagement and leaves readers unclear about the study's primary goals.
-Explicitly state the hypothesis and research objectives in both the abstract and the introduction. For example, include phrases such as: “This study hypothesizes that specific clinical and neurophysiological factors modulate PPT in SCI patients.” This will guide readers and improve the study’s framing.
Answer: Thank you. Following your suggestion, we have now incorporated this point into the introduction and abstract section of our manuscript.
- Results: The results section is overly text-heavy, making it difficult for readers to grasp the key findings and their significance. Incorporate graphical representations of the data to improve accessibility and interpretation. Suggested including plots showing correlations, such as PPT vs. age and PPT vs. motor strength.
Answer: Thank you for your valuable feedback regarding the presentation of the results section. We understand the importance of improving the accessibility and interpretation of key findings by incorporating graphical representations. In response, we have added supplementary material containing additional figures and visualizations that illustrate the results in a clearer and more concise manner. We appreciate your suggestion, which has helped us enhance the presentation and readability of the manuscript. Thank you for your thoughtful feedback.
- Discussion: The study notes a positive association between age and PPT, contrasting with trends observed in the general population, but does not provide a physiological explanation for this observation.
- Expand the discussion to include potential mechanisms underlying this finding. For example, consider discussing age-related changes in sensory processing, neuroplasticity, or central sensitization. Reference relevant literature to provide a more nuanced understanding of this counterintuitive result.
Answer: Thank you for your valuable comment regarding the association between age and PPT and the need for a more detailed explanation of the physiological mechanisms underlying this finding. In response, we have expanded the discussion section to include potential mechanisms that may explain this counterintuitive result. Specifically, we have incorporated an analysis of age-related changes in sensory processing, highlighting how the natural degeneration of small-diameter peripheral nerve fibers in older individuals can reduce nociceptive input and potentially increase pain thresholds. Additionally, we discuss how neuroplasticity in the context of SCI may lead to enhanced descending inhibitory control in older patients, serving as a compensatory mechanism to modulate pain perception. We also address the role of central sensitization, noting that its effects may stabilize or diminish with age in the SCI population, resulting in less reactive pain processing compared to younger individuals. Furthermore, we explore psychological and behavioral adaptations, such as the development of effective coping strategies in older patients, which may influence pain perception and contribute to the observed increase in PPT with age.
These additions are supported by references to relevant literature to provide a more nuanced understanding of this complex phenomenon. We believe this expanded discussion offers greater clarity and depth, addressing the reviewer's concern and enhancing the overall quality of the manuscript. Thank you for your thoughtful feedback, which has greatly improved our interpretation of the results.
- The relationship between anxiety and increased PPT is noted but not adequately explained. The discussion does not delve into how stress-induced analgesia might modulate sensory processing in SCI patients.
- Provide a detailed explanation of stress-induced analgesia, supported by references to mechanisms such as endogenous opioid release or alterations in central pain modulation pathways. Discuss how these mechanisms might differ in SCI patients compared to the general population.
Answer: Thank you for highlighting this important point. In the section of discussion, we explore more this topic and the mechanisms that are related.
Minor Points
- The abstract does not define the ‘DEFINE’ cohort, potentially confusing readers unfamiliar with the study.
Answer: Thank you. Following your suggestion, we have now incorporated this point into the abstract section of our manuscript.
- Methods Line 113: “Motor Strength Function function” includes a typographical error.
Answer: Thank you very much for your detailed and constructive feedback.
- Expand the discussion to connect findings to broader neurological and clinical contexts.
Answer: Thank you for your suggestion. This expanded discussion situates our findings within a broader neurological and clinical framework, demonstrating their relevance to the wider field of pain management and neurorehabilitation. We have incorporated this perspective into the revised manuscript and appreciate your feedback, which has helped enhance the context and significance of our results.
- Highlight the clinical implications of these findings by outlining how they could inform personalized pain management strategies.
Answer: Thank you for your insightful comment regarding the clinical implications of our findings. We have expanded the discussion to outline how these results could inform personalized pain management strategies for individuals with SCI. By integrating these individualized factors into pain management plans, clinicians can better address the unique pain experiences of SCI patients, leading to more effective and tailored interventions. We appreciate the opportunity to expand on the clinical relevance of our findings and have included this discussion in the revised manuscript. Thank you for your valuable feedback.
Reviewer 2 Report
Comments and Suggestions for Authors
Healthcare – 3410776
“Role of maladaptive plasticity in modulating pain pressure threshold post-spinal cord injury.
This study by Imamura et al. explores the complex interplay of several factors influencing Pressure Pain Threshold (PPT) in individuals with spinal cord injury (SCI), providing data into pain sensitization mechanisms and their potential implications for personalized rehabilitation strategies. However, the study lacks a control group, and several aspects remain insufficiently explained:
(1) The sample is predominantly male, which may not adequately reflect the experiences of female SCI patients. How might this gender imbalance affect the generalizability of the results, and were gender differences in pain sensitivity specifically analyzed? The authors should discuss this concern.
(2) The manuscript would benefit from a more detailed spectral characterization of frequency power oscillations, rather than relying solely on average values presented in a table. Why was the low-beta frequency band specifically identified as a predictor of PPT, and what mechanisms explain its association with increased pain sensitivity in SCI patients?
(3) Regarding PPT measurements, this study did not present a control group, which is particularly important for this type of mechanical comparative analysis. Additionally, were any adjustments or calibrations made to address potential inconsistencies in PPT measurements caused by variations in the algometer application technique across participants?
(4) The study mentions the inability to examine changes over time due to its cross-sectional design. How might the temporal dynamics of pain perception and sensitivity, such as pain adaptation or delayed onset of neuropathic pain, influence the findings?
(5) The association between anxiety and increased PPT is counterintuitive to established literature. What factors or coping mechanisms unique to the SCI population might explain this unexpected relationship?
(6) Since PPT was measured solely in the thenar region, how generalizable are the findings to other body areas that may exhibit different pain sensitivities based on injury levels or sensory innervation? Would testing additional sites offer a more comprehensive understanding?
(7) The results section references "supplementary material," yet these data were not provided for review.
Author Response
REVIEWER 02:
This study by Imamura et al. explores the complex interplay of several factors influencing Pressure Pain Threshold (PPT) in individuals with spinal cord injury (SCI), providing data into pain sensitization mechanisms and their potential implications for personalized rehabilitation strategies. However, the study lacks a control group, and several aspects remain insufficiently explained:
(1) The sample is predominantly male, which may not adequately reflect the experiences of female SCI patients. How might this gender imbalance affect the generalizability of the results, and were gender differences in pain sensitivity specifically analyzed? The authors should discuss this concern.
Answer: Thank you for your insightful comment regarding the gender imbalance in our sample. We acknowledge that the sample is predominantly male, which reflects the real-world epidemiology of spinal cord injury (SCI), as males account for the majority of SCI cases globally. This demographic distribution has been consistently reported in the literature and is a limitation inherent to the condition rather than a sampling bias. We agree that the predominance of male participants may affect the generalizability of our findings, particularly when considering potential gender differences in pain sensitivity. While gender differences in pain sensitivity were not specifically analyzed in this study due to the small number of female participants, we recognize that women may exhibit different pain thresholds or experiences of pain due to biological and psychosocial factors. This is an important limitation and we have emphasized this in the “limitations section” of the manuscript. Thank you again for your valuable feedback, which has helped improve our manuscript.
(2) The manuscript would benefit from a more detailed spectral characterization of frequency power oscillations, rather than relying solely on average values presented in a table. Why was the low-beta frequency band specifically identified as a predictor of PPT, and what mechanisms explain its association with increased pain sensitivity in SCI patients?
Answer: Thank you for your comment regarding the spectral characterization of frequency power oscillations. We agree that a more detailed spectral analysis could provide additional depth to the findings and improve the interpretability of the frequency power oscillations. However, in this study, we focused on presenting the average values to provide a concise summary and ensure clarity in the context of our primary objective. Future work will aim to expand on these findings with a more detailed spectral analysis. The low-beta frequency band was specifically identified as a predictor of PPT due to its consistent association with pain processing and sensory modulation observed in previous studies. Low-beta oscillations are thought to reflect cortical processing of somatosensory inputs and the integration of nociceptive information. In SCI patients, maladaptive plasticity within cortical and subcortical structures may lead to altered beta activity, which has been linked to increased pain sensitivity and impaired descending pain inhibition. This aligns with findings that suggest a role for low-beta activity in maintaining hyperexcitability within sensory networks, contributing to the perception of pain.
We appreciate your suggestion to enhance the spectral characterization and will explore this avenue in future studies to further elucidate the role of frequency power oscillations in pain modulation. Thank you for your valuable feedback, which has been instrumental in refining the manuscript.
(3) Regarding PPT measurements, this study did not present a control group, which is particularly important for this type of mechanical comparative analysis. Additionally, were any adjustments or calibrations made to address potential inconsistencies in PPT measurements caused by variations in the algometer application technique across participants?
Answer: Thank you for your insightful comment regarding the absence of a control group and potential inconsistencies in PPT measurements. We acknowledge that including a control group would provide a valuable point of reference for interpreting the results, particularly in comparative mechanical analyses. However, our study focused on exploring pain thresholds within the SCI population, and future research incorporating a healthy control group would undoubtedly enhance the generalizability and robustness of these findings.
Regarding potential variations in algometer application technique, we ensured that rigorous calibration procedures were followed. Specifically, the algometer was calibrated before each assessment session when the device was turned on to ensure consistent pressure application across participants. Additionally, all assessments were conducted by trained personnel following a standardized protocol to minimize inter-rater variability and ensure uniform application techniques. We sincerely appreciate your feedback, which has helped us strengthen the presentation of our methods.
(4) The study mentions the inability to examine changes over time due to its cross-sectional design. How might the temporal dynamics of pain perception and sensitivity, such as pain adaptation or delayed onset of neuropathic pain, influence the findings?
Answer: Thank you for your thoughtful comment regarding the limitations of the cross-sectional design and its implications for understanding the temporal dynamics of pain perception in individuals with SCI. We fully agree that processes such as pain adaptation and the delayed onset of neuropathic pain are critical aspects that could influence pain sensitivity and perception over time. In the limitations section of our manuscript, we have acknowledged this as a key limitation, noting that our study design does not allow for an in-depth exploration of changes in pain sensitivity over time.
We also emphasize that a longitudinal approach would be better suited to examine these dynamics and provide insights into the onset and progression of neuropathic pain. However, we chose a cross-sectional design to reduce the potential for participant dropout, which can pose significant challenges in longitudinal studies, and to effectively assess the prevalence of altered PPT within the SCI population at a single time point. To further address your valuable suggestion, we will refine the limitation to explicitly highlight the importance of longitudinal studies in complementing our findings and advancing the understanding of the temporal aspects of pain perception in SCI. We greatly appreciate your feedback and its role in enhancing the clarity and completeness of our manuscript.
(5) The association between anxiety and increased PPT is counterintuitive to established literature. What factors or coping mechanisms unique to the SCI population might explain this unexpected relationship?
Answer: Thank you for your comment regarding the counterintuitive association between anxiety and increased PPT in our study. We agree that these finding challenges established literature, which typically links anxiety with decreased pain tolerance due to heightened perception of pain and distress. In our study, we propose that this unique relationship in the SCI population may be explained by several factors and coping mechanisms specific to this group. Spinal cord injury often results in significant neuroplastic changes, particularly within the autonomic nervous system and central pain modulation pathways. These changes may lead to altered sensory processing, where anxiety triggers stress-induced analgesia (SIA). SIA is a well-documented phenomenon in which exposure to stressors, including anxiety, activates the body’s endogenous pain inhibition systems. This process involves the release of endogenous opioids, norepinephrine, and serotonin, which act on descending pain pathways to suppress pain perception.
In SCI patients, these pathways may be further modified due to the disruption of normal nociceptive signaling and increased reliance on descending inhibitory control as a compensatory mechanism. This adaptation could explain the atypical response of elevated PPT in individuals with higher anxiety levels, as it reflects the activation of SIA in response to chronic pain or stress. Additionally, SCI patients may develop unique psychological coping strategies over time, which could influence this relationship. Anxiety, in this context, might paradoxically enhance pain tolerance as part of a survival mechanism against persistent discomfort. For example, chronic exposure to stressors may condition the activation of endogenous analgesic pathways, resulting in a temporary increase in PPT during stressful situations. We appreciate this opportunity to expand on our findings and will incorporate a detailed discussion of these mechanisms in the discussion section.
(6) Since PPT was measured solely in the thenar region, how generalizable are the findings to other body areas that may exhibit different pain sensitivities based on injury levels or sensory innervation? Would testing additional sites offer a more comprehensive understanding?
Answer: Thank you for your thoughtful comment on the generalizability of PPT measurements taken solely in the thenar region. While we acknowledge that testing additional body sites could provide a more comprehensive understanding of pain sensitivity patterns in SCI patients, our findings suggest that the characteristics of the injury itself—such as its level, type, and severity—play a more substantial role in shaping pain sensitivity than the specific site of measurement. Specifically, our results indicate that PPT in SCI patients is significantly influenced by the nature and severity of the injury, as classified by the American Spinal Injury Association (AIS) scale. Patients with incomplete SCIs (AIS grades B through E) consistently exhibit lower PPTs, reflecting heightened pain sensitivity. This increased sensitivity may be attributed to the interaction between intact and damaged neural circuits, where partial preservation of nerve pathways contributes to altered pain processing and amplified responses. These findings align with previous studies reporting similar phenomena in patients with incomplete injuries.
Additionally, our analysis revealed that the etiology of SCI—traumatic versus non-traumatic—further influences PPT outcomes. Non-traumatic SCIs are associated with significantly lower PPTs, likely due to the chronic and progressive nature of these injuries, which involve prolonged nerve irritation and maladaptive neural changes such as central sensitization. These findings are consistent with prior research highlighting the heightened pain sensitivity in non-traumatic SCI cases. Furthermore, we conducted an additional analysis comparing PPT measurements in patients with cervical injuries versus those with non-cervical injuries to evaluate whether injury level influenced our results. Importantly, this analysis demonstrated no significant differences, suggesting that the findings in the thenar region are robust across varying injury levels.
While the thenar region provides a reliable and consistent measurement site, these results support the notion that the characteristics of the SCI itself—rather than the specific body site measured—are the primary determinants of pain sensitivity. Nonetheless, we agree that testing additional sites could enhance understanding of spatial variability in pain sensitivity, and we recommend this as a direction for future research. We appreciate your feedback and have expanded the manuscript to address these points more explicitly, ensuring greater clarity and depth in discussing our findings.
(7) The results section references "supplementary material," yet these data were not provided for review.
Answer: Thank you for bringing this to our attention. We apologize for the oversight regarding the supplementary material. The referenced supplementary data will be included in the revised submission to ensure it is available for review. These materials contain additional details that support our findings, including extended analyses and relevant data tables, which provide further transparency and depth to the results.
We appreciate your understanding, and we will make sure this is appropriately addressed in the revised manuscript. Thank you for highlighting this important point.
Round 2
Reviewer 2 Report
Comments and Suggestions for Authors
The authors have elegantly addressed all my concerns about the study.
I am satisfied.
Congratulations.
Author Response
Thank you for your kind words and positive feedback on our study. We truly appreciate your thoughtful review and support throughout this process.